# Insights into Biological and Ecological Features of Four Rare and Endemic Plants from the Northern Tian Shan (Kazakhstan)

**DOI:** 10.3390/plants14152305

**Published:** 2025-07-26

**Authors:** Gulbanu Sadyrova, Aisha Taskuzhina, Alexandr Pozharskiy, Kuralai Orazbekova, Kirill Yanin, Nazym Kerimbek, Saule Zhamilova, Gulzhanat Kamiyeva, Ainur Tanybaeva, Dilyara Gritsenko

**Affiliations:** 1Faculty of Geography and Environmental Management, Al-Farabi Kazakh National University, Almaty 050040, Kazakhstan; sadyrova.gulbanu@kaznu.kz (G.S.); ainur.tanybaeva@kaznu.kz (A.T.); 2Laboratory of Molecular Biology, Institute of Plant Biology and Biotechnology, Almaty 050040, Kazakhstan; ataskuzina@gmail.com (A.T.); aspozharsky@gmail.com (A.P.); yanin.kirill97@gmail.com (K.Y.); nazym_kerimbek@mail.ru (N.K.); 3Faculty of Biology and Biotechnology, Al-Farabi Kazakh National University, Almaty 050040, Kazakhstan; 4Research Center AgriBioTech, Almaty 050040, Kazakhstan; 5Institute of Geography and Water Security, Almaty 050010, Kazakhstan; kuralay_orazbekova@mail.ru; 6Faculty of Natural Sciences and Geography, Abai Kazakh National Pedagogical University, Almaty 050010, Kazakhstan; dsm1750@gmail.com (S.Z.); kamievags@mail.ru (G.K.)

**Keywords:** threatened species, chloroplast genome, *Taraxacum kok-saghyz*, *Astragalus rubtzovii*, *Schmalhausenia nidulans*, *Rheum wittrockii*

## Abstract

This study presents an integrative investigation of four rare and threatened plant species—*Taraxacum kok-saghyz* L.E. Rodin, *Astragalus rubtzovii* Boriss., *Schmalhausenia nidulans* (Regel) Petr., and *Rheum wittrockii* Lundstr.—native to the Ile Alatau and Ketmen ridges of the Northern Tian Shan in Kazakhstan. Combining chloroplast genome sequencing, geobotanical surveys, and anatomical and population structure analyses, we aimed to assess the ecological adaptation, genetic distinctiveness, and conservation status of these species. Field surveys revealed that population structures varied across species, with *T. kok-saghyz* and *S. nidulans* dominated by mature vegetative and generative individuals, while *A. rubtzovii* and *R. wittrockii* exhibited stable age spectra marked by reproductive maturity and ongoing recruitment. Chloroplast genome assemblies revealed characteristic patterns of plastid evolution, including structural conservation in *S. nidulans* and *R. wittrockii*, and a reduced inverted repeat region in *A. rubtzovii*, consistent with its placement in the IR-lacking clade of Fabaceae. Morphological and anatomical traits reflected habitat-specific adaptations such as tomentose surfaces, thickened epidermis, and efficient vascular systems. Despite these adaptations, anthropogenic pressures including overgrazing and habitat degradation pose significant risks to population viability. Our findings underscore the need for targeted conservation measures, continuous monitoring, and habitat management to ensure the long-term survival of these ecologically and genetically valuable endemic species.

## 1. Introduction

Kazakhstan, the largest landlocked country in the world, possesses an incredibly diverse range of plant species and is home to over 6000 species of plants, with a significant proportion being endemic to the region, including 451 taxa that constitute 7.97% of the total vascular plant diversity, primarily concentrated in the Northern Tian Shan and Dzungarian Alatau [1]. Among them, 387 species are listed in the *Red Book of Kazakhstan* as rare and threatened [2]. In the context of accelerating global climate change, the preservation of plant species diversity has emerged as a critical priority for ensuring ecological resilience and mitigating the ongoing loss of biodiversity [3]. Biodiversity loss can result in profound ecological consequences, including ecosystem degradation and the depletion of essential natural resources that support human livelihoods [4,5,6,7]. Effective conservation of rare and threatened plant species requires the development and implementation of comprehensive strategies. These include the establishment of protected areas, restoration of native habitats, and the advancement of targeted scientific research. Furthermore, increasing public awareness and actively engaging local communities in conservation efforts are essential components for the long-term success of biodiversity preservation initiatives [8,9,10,11,12,13]. In recent decades, the natural ecosystems of the Northern Tian Shan, particularly the Ketmen (Uzynkara) and Ile Alatau ridges, have experienced significant anthropogenic impact [14]. These regions serve as valuable model systems for the assessment of the dynamics of plant populations under varying ecological conditions. The pronounced altitudinal gradient and habitat heterogeneity of the Ketmen and Ile Alatau ridges provide a unique opportunity to investigate the structural and functional characteristics of populations, as well as their reproductive strategies and adaptive responses to environmental stressors. Such studies are essential for developing effective conservation measures, including habitat management, restoration strategies, and the identification of microrefugia that support long-term species persistence amid ongoing climatic and anthropogenic pressures. This study focused on natural populations of rare and threatened plant species endemic to the Northern Tien Shan region of Kazakhstan, representing three families and four genera. These include taproot species *Taraxacum kok-saghyz* L.E. Rodin (NCBI:txid333970, Asteraceae), *Schmalhausenia nidulans* (Regel) Petr. (synonym of *Arctium eriophorum* (Regel & Schmalh.) Kuntze, 1891; NCBI:txid254703; Asteraceae), and *Astragalus rubtzovii* Boriss. (NCBI:txid3084330; Fabaceae), as well as short-rhizome species *Rheum wittrockii* Lundstr. (NCBI:txid137228; Polygonaceae). *T. kok-saghyz*, commonly known as kok-saghyz (from Kazakh, meaning ‘green gum’), or Russian dandelion, is an important species of Kazakhstan and a valuable source of natural rubber [15]. *A. rubtzovii* Boriss. is a narrow-range xeromesophytic species restricted to the relict dune sands of Kum-Tekey in southeastern Kazakhstan, where it demonstrates ecological specialization to sandy substrates [16]. *S. nidulans* is a high-mountain perennial species occurring in the alpine and subalpine belts, where it inhabits fine- and coarse-textured moist soils on steep mountain slopes and within relict glacial valleys of the Ile, Ketpen, and Kyrgyz Alatau ranges [17]. *R. wittrockii* is a relict perennial species confined to the Northern Tian Shan, occurring in rocky riverbeds, stony and grassy slopes, and shrub-dominated habitats, where it is well adapted to arid, high-altitude environments; it is of notable importance for its nutritional, pharmacological, and ornamental properties [18]. Despite their ecological significance and endemism, the studied species face increasing threats due to anthropogenic pressures and are listed in the *Red Book of Kazakhstan* [2]. Overgrazing, land conversion for agriculture, and other forms of human disturbance near settlements contribute significantly to habitat degradation and population decline [19]. In particular, the quantitative abundance of these species is decreasing as a result of unsustainable land use practices such as plowing and intensive livestock grazing. These impacts are especially critical for narrowly distributed taxa like *T. kok-saghyz* and *A. rubtzovii*, whose populations are already fragmented and highly specialized to unique edaphic conditions. Without the implementation of targeted conservation measures, including habitat protection and monitoring of population dynamics, the long-term persistence of these rare and endangered species remains at risk.

In this study, we conducted the first comprehensive investigation combining chloroplast genome analysis with geobotanical characterization, population structure assessment, and anatomical studies of rare and threatened plant species in the Northern Tian Shan, providing new insights into their ecological adaptation, genetic distinctiveness, and conservation status.

## 2. Results

### 2.1. Ecological, Anatomical, and Genomic Features of Taraxacum kok-saghyz

#### 2.1.1. Habitat and Population Structure of *T. kok-saghyz*

*T. kok-saghyz* occurs in the intermountain valleys of Kegen and Saryzhaz in the Almaty region, growing in sharply continental mountainous climates of the Northern Tian Shan (Figure 1a). It is a perennial mesophytic herb with a high moisture requirement and the capacity to enter dormancy during drought conditions. The species exhibits a fragmented distribution across its habitat, with total projective cover ranging from 20% to 30%. It typically forms small, scattered populations in high-altitude environments between 1800 and 2000 m above sea level, preferring meadows, gravelly slopes, river valleys, and moist ravines on northern mountain exposures. Age structure analysis revealed a predominance of virginile (mature vegetative) individuals, comprising up to 86% of the populations. Juvenile and immature plants were present in moderate numbers, while old generative, subsenile, and senile individuals were observed only sporadically, indicating limited progression to later developmental stages.

#### 2.1.2. Morphological and Anatomical Characteristics of *T. kok-saghyz*

The taproot penetrates deeply into the soil and is covered with dark bark, beneath which elastic, rubber-containing threads become visible when the outer cork layer is removed. Individuals were irregularly distributed within 100 m^2^ plots, with rosette diameters averaging 15–20 cm (Figure 1b). Leaves were entirely or shallowly serrated, reaching up to 10 cm in length and 3 cm in width. Morphometric data across age groups and populations are summarized in Table 1.

Cross-sectional analysis revealed that the stem consists of three distinct zones: a single-layered epidermis with thickened walls and cuticle, a six-to-seven-layered parenchymatous cortex (8.97 ± 0.8 μm), and a central vascular ring (Figure 1c). Vascular bundles were surrounded by 3–4 layers of sclerenchyma (11.8 ± 0.4 μm), with wider xylem elements than phloem. The leaf displayed dorsoventral anatomy, with a single-layered upper and lower epidermis (6.07 ± 0.41 μm and 4.89 ± 0.31 μm). The mesophyll was differentiated into columnar and spongy layers, with the latter containing prominent intercellular spaces indicative of adaptation to humid conditions.

#### 2.1.3. Chloroplast Genome Analysis of *T. kok-saghyz*

The chloroplast genome of *T. kok-saghyz* was assembled as a circular sequence of 151,353 bp with a GC content of 37.68% (Figure 1d). The genome contains 109 unique genes, including 88 protein-coding genes, 8 rRNA genes, and 33 tRNA genes. The functional analysis of protein-coding genes revealed 7 genes encoding components of PSI, 14 for PSII, 6 for ATP synthase, 11 for NADH dehydrogenase, and 6 for the cytochrome b6/f complex. We also identified genes encoding ribosomal proteins (22); RNA polymerase components (4); and other functional genes, including *rbcL*, *accD*, *ccsA*, *cemA*, *clpP1*, *infA*, *matK*, *pafI*, *pafII*, *pbf1*, *ycf1*, and *ycf2*. Similar to *S. nidulans*, *T. kok-saghyz* exhibited 17 duplicated genes characteristic of well-preserved inverted repeat regions typically found in most angiosperm chloroplast genomes. The plastid phylogeny confirmed the placement of *T. kok-saghyz* within the *Taraxacum* clade with high bootstrap support (Figure 1e); however, it was positioned apart of other *Taraxcum* species, including the available *T. kok-saghyz* reference (NC_032057).

### 2.2. Ecological, Anatomical, and Genomic Features of Astragalus rubtzovii

#### 2.2.1. Habitat and Population Structure of *A. rubtzovii*

*A. rubtzovii* is a perennial, polycarpic, xeromesophytic psammophyte restricted to relict dune sands with sufficient moisture (Figure 2a). It occurs exclusively in the sandy massif of Kum-Tekey on the Uzynkara Ridge (Kegen District), occupying an area of approximately 15 km^2^ at elevations ranging from 1800 to 2000 masl. Field observations indicate that the species forms sparse but uniform stands, with an overall projective cover of 45–70% and dominant species reaching heights of 80–90 cm. The vegetation is dominated by sod-forming grasses such as *Leymus racemosus*, *Stipa capillata*, *Festuca ganeschini*, and *F. valesiaca*, accompanied by a diversity of herbs, including *Medicago falcata*, *Artemisia sieversiana*, *Lappula microcarpa*, and multiple Astragalus species.

Population structure analysis revealed a predominance of middle-aged and young generative individuals, comprising approximately 83% of the total population, indicating demographic stability. Young virginile individuals were well represented, while old generative, subsenile, and senile individuals were rare or absent.

#### 2.2.2. Morphological and Anatomical Characteristics of *A. rubtzovii*

*A. rubtzovii* reaches a height of 12–40 cm and is distinguished by erect stems; triangular–lanceolate stipules with white, woolly margins; and squarrose, hairy leaves measuring 10–30 cm in length. The leaves bear 15–18 pairs of lanceolate–oblong leaflets. The inflorescence is a loosely arranged 8–12 cm long raceme with yellow corollas and leathery, oblong pods. Morphometric analysis revealed that in the immature stage, individuals reach a height of 7–10 cm and produce 3–4 pairs of leaflets (Table 2). During the virginile stage, plants grow to 10–15 cm and begin to exhibit traits typical of mature individuals. Young generative plants form reproductive structures, including shortened inflorescences, and bear leaves with 7–10 pairs of leaflets.

In cross-section, *A. rubtzovii* leaves have smooth margins, well-defined polygonal upper epidermal cells (4.68 ± 0.8 μm thick), and slightly curved lower epidermal cells (3.66 ± 0.7 μm) (Figure 2b). Anomocytic stomata and both unicellular and bicellular trichomes are present on both leaf surfaces. The mesophyll is differentiated into columnar (10.8 ± 1.2 μm) and spongy (14.6 ± 0.61 μm) layers, with vascular bundles clearly developed in the leaf veins. Chloroplasts are abundant in the mesophyll. The root exhibits a well-developed secondary structure with a distinct periderm, secondary parenchyma, and central cylinder. Primary parenchyma cells measure 7.09 ± 1.87 μm, while sclerenchyma cells are 6.31 ± 0.5 μm thick and arranged in three to four rows. The central cylinder contains 14–16 collateral vascular bundles, with xylem elements displaying a radial arrangement and varying diameters. The pericycle is clearly differentiated, and the endodermis is well defined.

#### 2.2.3. Chloroplast Genome Analysis of *A. rubtzovii*

The complete chloroplast genome of *A. rubtzovii* was assembled into a circular sequence of 122,984 bp with a GC content of 34.15% (Figure 2c). We identified a total of 110 unique genes, including 81 protein-coding genes, 7 ribosomal RNA (rRNA) genes, and 27 transfer RNA (tRNA) genes. The functional categorization of protein-coding genes revealed 7 genes encoding subunits of photosystem I (PSI), 14 for photosystem II (PSII), 6 for ATP synthase, 11 for NADH dehydrogenase, and 6 for the cytochrome b6/f complex. We also identified a complete set of ribosomal protein genes (22); RNA polymerase genes (4); and other genes, including *rbcL*, *accD*, *ccsA*, *cemA*, *clpP1*, *matK*, *pafI*, *pafII*, *pbf1*, *ycf1*, and *ycf2*. Notable among our findings was the identification of only three duplicated genes (*rrn23-fragment*, *trnM-CAU*, and *trnV-GAC*), consistent with the typical loss of one inverted repeat in members of the Inverted Repeat-Lacking Clade (IRLC) of Fabaceae. Unlike typical angiosperm chloroplasts, the *A. rubtzovii* chloroplast lacks the standard quadripartite structure with large, inverted repeat regions, while still maintaining all essential chloroplast genes for photosynthetic function. The phylogenetic analysis placed *A. rubtzovii* firmly within the clade corresponding to the *Astragalus* genus with 100% bootstrap support; the closest identified species were *A. flexus*, *A. vulpinus*, and *A. sieversianus* (Figure 2d). The overall tree demonstrated clustering of the species, complying with the genera, with the exception of *A. complanatus* (NC_065023), which was grouped along with the single species of genera *Carmichaelia*, *Sphaerophysa*, and *Lessertia*.

### 2.3. Ecological, Anatomical, and Genomic Features of Schmalhausenia nidulans

#### 2.3.1. Habitat and Population Structure of *S. nidulans*

*S. nidulans* is a perennial herbaceous species inhabiting ancient glacial valleys and alpine belts of the Ile Alatau and Uzynkara ridges (Figure 3a). It predominantly grows on rubble–fine earth lawns at high elevations, ranging from 2200 to 3340 masl. Individuals are sparsely and irregularly distributed, often found as solitary plants.

Population analysis revealed a predominance of generative individuals, with a significant number of virginile plants, suggesting stable coenopopulation dynamics. Most populations fall under the category of young or mature normal types. However, in areas subjected to overgrazing and trampling, population viability declines, emphasizing the need for moderate land use and targeted conservation measures.

#### 2.3.2. Morphological and Anatomical Characteristics of *S. nidulans*

The species is characterized by a single, erect stem with a cobwebby–tomentose surface, reaching up to 50 cm in height. The stem base retains dark remnants of dead leaves and is densely foliated. Leaves are oblong–lanceolate and densely covered with gray tomentum on both sides, contributing to reduced water loss under alpine conditions. Inflorescences are large, solitary capitula measuring 4–5 cm in diameter and are also tomentose, reflecting adaptation to high-elevation, UV-intensive environments. Morphometric measurements of leaf length and plant height across developmental stages in two populations are provided in Table 3.

The stem anatomy exhibits a secondary structure consistent with Asteraceae (Figure 3b). The epidermis is composed of oval or round cells filled with loose parenchyma (7.31 ± 0.51 μm). The cortex features oval to rectangular parenchymatous cells forming uni- or multicellular trichomes. Primary-bundle sheath parenchyma cells are thick-walled (5.29 ± 1.13 μm). The central cylinder includes eight to nine elliptical vascular bundles enclosed by sclerenchyma. Cambial cells are located between the xylem and phloem, and sclerenchyma fibers are thickened on both sides of the phloem. The stem shows a radial anatomical organization typical of a ray-type structure.

#### 2.3.3. Chloroplast Genome Analysis of *S. nidulans*

The chloroplast genome of *S. nidulans* was assembled as a circular sequence of 152,659 bp with a GC content of 37.68% (Figure 3c). We annotated 110 unique genes, including 89 protein-coding genes, 8 rRNA genes, and 31 tRNA genes. Among the protein-coding genes, we identified 7 genes for PSI, 14 for PSII, 6 for ATP synthase, 11 for NADH dehydrogenase, and 6 for the cytochrome b6/f complex. The genome also contains genes encoding ribosomal proteins (22); RNA polymerase components (4); and other functional proteins, such as *rbcL*, *accD*, *ccsA*, *cemA*, *clpP1*, *infA*, *matK*, *pafI*, *pafII*, *pbf1*, *ycf1*, *ycf1-fragment*, and *ycf2*. A total of 16 duplicated genes were identified, including key genes typically found in inverted repeat regions, such as *ndhB*, *rpl2*, *rpl23*, *rps7*, *rps19*, *rrn16*, *rrn23*, *rrn4.5*, *rrn5*, and several tRNA genes. This pattern suggests well-preserved, inverted repeat structures in this species. The phylogenetic analysis resulted in *S. nidulans* (syn. *Arctium eriophorum*) being clustered closely with the only available species of *Arctium*, *A. lappa*, with 100% bootstrap support (Figure 3d). This small *Arctium* sub-clade was clustered along with the species of *Saussurea* with high bootstrap support

### 2.4. Ecological, Anatomical, and Genomic Features of Rheum wittrockii

#### 2.4.1. Habitat and Population Structure of *R. wittrockii*

*R. wittrockii* is a polycarpic herbaceous plant occurring on grassy and forested mountain slopes, reaching into the subalpine belts of the Ile Alatau and Uzynkara ridges at elevations ranging from 2000 to 3300 m above sea level (Figure 4a). It grows within diverse plant communities, accompanied by arboreal species such as *Sorbus tianschanica*, *Populus tremula*, and *Betula tianshanica*; shrub species including *Lonicera karelinii*, *L. hispida*, *Rosa laxa*, and *Spiraea lasiocarpa*; and herbaceous species such as *Geranium collinum*, *Ligularia macrophylla*, *Rumex tianschanicus*, *Codonopis clematidae*, *Phlomis oreophylla*, *Veratrum lobelianum*, and *Heracleum dissectum*.

Population assessments indicate that *R. wittrockii* coenopopulations are demographically stable, supported by a predominance of virginile and middle-aged generative individuals and regular recruitment of new seedlings. Virginile individuals demonstrated the highest size values in population 1 and the lowest values in population 2, whereas the number of leaves remained relatively consistent. In population 3, only one middle-aged and one old generative plant were recorded; therefore, their measurements were excluded from statistical evaluation.

#### 2.4.2. Morphological and Anatomical Characteristics of *R. wittrockii*

*R. wittrockii* is a tall, herbaceous perennial that reaches up to 1 m in height and is easily recognized by its thick vertical rhizome and erect, finely grooved stem. The leaves are large, broadly triangular to ovoid in shape, and can reach dimensions of up to 40 cm in length and 30 cm in width. The inflorescence is a rare, spreading panicle borne on a stem measuring 50–100 cm in height. Morphometric data across developmental stages in two populations of *R. wittrockii* are summarized in Table 4.

The leaf anatomy of *R. wittrockii* demonstrates clear adaptation to mesophytic conditions (Figure 4b). It features a distinct upper and lower epidermis with well-developed unicellular trichomes. The upper epidermal cells are convex, elongated, and thick-walled (7.38 ± 1.03 μm), while the lower epidermis is thinner (6.01 ± 1.16 μm) and displays a high density of stomata typical of dicotyledonous species. The leaf blade reaches approximately 1000 μm in thickness. The mesophyll is differentiated into irregularly shaped loose parenchyma cells (12.1 ± 0.87 μm) interspersed with prominent intercellular spaces. A central, enlarged collateral vascular bundle is flanked by three to five smaller bundles, all surrounded by sclerenchyma. Beneath these bundles, idioblasts with fluid-filled lumens are observed.

In the transverse section, the stem is slightly ribbed and covered by a single-layer epidermis bearing simple and capitate hairs. Epidermal cells are thick-walled (11.6 ± 0.81 μm) and coated with a cuticle. Angular collenchyma alternates with two to three layers of chlorenchyma beneath the epidermis. A clearly defined endodermal layer (11.97 ± 0.7 μm) encases the central cylinder. The vascular system consists of 12–14 open collateral bundles of varying size, each reinforced by several layers of sclerenchyma (9.17 ± 1.6 μm). Phloem and xylem tissues are well organized, with cambial cells present between them. The pith is composed of large, rounded parenchyma cells interspersed with idioblasts.

#### 2.4.3. Chloroplast Genome Analysis of *R. wittrockii*

The chloroplast genome of *R. wittrockii* was assembled as a circular sequence with a length of 141,379 bp and a GC content of 36.83% (Figure 4c). The genome contains 110 unique genes, comprising 85 protein-coding genes, 7 rRNA genes, and 27 tRNA genes. The protein-coding genes include 7 genes for PSI, 15 for PSII, 6 for ATP synthase, 11 for NADH dehydrogenase, and 6 for the cytochrome b6/f complex. Additionally, we identified genes for ribosomal proteins (21); RNA polymerase (4); and other functional genes, including *rbcL*, *accD*, *ccsA*, *cemA*, *clpP1*, *infA*, *matK*, *pafI*, *pafII*, *pbf1*, *ycf1*, *and ycf2*. Eight detected genes (*psbA*, *rps12*, *rrn4.5*, *rrn5*, *trnM-CAU*, *trnN-GUU*, *trnR-ACG*, and *ycf1*) have been found to have duplicates; however, the obtained plastid genome assembly does not provided evidence of the inverted repeat regions typical for angiosperm plastids. The phylogenetic analysis of *R. wittrockii* and other species of the Polygonideae subfamily revealed clustering of *R. wittrockii*, along with other *Rheum* species; the closest species were *R. alexandrae* and *R. rhabarbarum*, with high bootstrap support (Figure 4d).

## 3. Discussion

Endemic plant species play an indispensable role in biodiversity, ecological balance, cultural heritage, and medicinal applications, uniquely contributing to the ecological identity of their native habitats through distinct local adaptations that enhance ecosystem resilience and functionality. However, despite their ecological and cultural significance, endemic species are particularly vulnerable due to their limited geographic distribution, small population sizes, and heightened sensitivity to fluctuations and anthropogenic pressures [20,21,22,23,24,25]. This study provides the first integrative ecological and genomic assessment of four rare endemic plant species in the Northern Tian Shan, offering valuable insights into their evolutionary distinctiveness, anatomical adaptations, and conservation requirements. Although chloroplast genomes are generally conserved in structure and gene content across angiosperms, they contain variable regions, particularly non-coding spacers, that provide useful markers for delineating conservation units, tracking maternal lineages, and detecting hybridization, supporting the management of genetic integrity and evolutionary potential in threatened taxa [26,27,28]. Notably, these species have been poorly studied or remain largely undocumented within the territory of Kazakhstan. Previous research efforts have primarily focused on *T. kok-saghyz*, a rubber-producing species of significant industrial interest, while other narrowly distributed taxa have received limited scientific attention [15,29,30,31,32]. The lack of prior comprehensive research underscores a critical knowledge gap in national biodiversity assessments, particularly regarding narrowly distributed taxa in mountainous ecosystems. By combining morphological, ecological, and genomic data, our findings contribute to a deeper understanding of the adaptive traits and conservation priorities of these endemic species.

### 3.1. T. kok-saghyz

The study of *T. kok-saghyz* reveals a fragmented population structure, with a predominance of virginile individuals and a low proportion of mature generative and senile plants. This skewed age distribution suggests challenges in successful generative recruitment and raises concerns about long-term population viability, especially under sustained anthropogenic pressures such as overgrazing and habitat disturbance. Anatomical adaptations such as elastic, rubber-containing root tissues; a thick epidermis; and a spongy mesophyll indicate mesophytic ecological preferences and physiological strategies suited to moist, high-altitude habitats. In contrast, the dandelion *Taraxacum officinale*—a closely related and ecologically widespread species—exhibits anatomical features that reflect its adaptability to a wide range of environmental conditions, including disturbed and drier habitats [33,34]. Its leaves possess a thinner epidermis and a less developed, spongy mesophyll compared to *T. kok-saghyz*, indicating a more generalized ecological strategy [35]. The root system of *T. officinale* lacks specialized rubber-containing tissues, aligning with its broader ecological tolerance and widespread distribution. These anatomical differences underscore the specialized adaptations of *T. kok-saghyz* to mesophytic, high-altitude environments, in contrast to the generalist nature of *T. officinale*.

The chloroplast genome of *T. kok-saghyz* (151,353 bp) we assembled aligns with previously published data [36], sharing the typical angiosperm quadripartite structure, Asteraceae-characteristic inversions, GC content (37.7%), and identical gene complement (134 genes). Notable features include three copies of the trnF-GAA gene (versus two in *T. officinale*) and the highest sequence variation in the accD gene and several intergenic regions. Surprisingly, as was revealed by the phylogenetic analysis, the obtained plastid genome assembly not only was different from the previously published *T. kok-saghyz* sequence (NC_032057) but was also clearly distinguished from other *Taraxacum* species. This fact raises the necessity of further studies on the structure of chloroplast and nuclear genomes to clarify whether the observed difference resulted from a sequencing and/or assembly mistake or if it indicates the existence of a previously unknown cryptic species of *Taraxacum.* The obtained complete chloroplast genome enables practical applications such as the development of species-specific markers to differentiate *T. kok-saghyz* from weedy relatives like *T. officinale* and *T. brevicorniculatum* for germplasm purification [37], the design of chloroplast transformation vectors with homologous flanking sequences for genetic engineering to enhance rubber production [38], and the identification of highly variable regions for the development of barcoding primers useful in tracking maternal lineage in potential hybridization events [36,39].

### 3.2. A. rubtzovii

*A. rubtzovii* demonstrates a demographically stable population structure dominated by middle-aged and young generative individuals. Its ecological specialization to relict dune sands in the Kum-Tekey massif is supported by extensive morphological adaptations such as trichome-rich leaf surfaces, thick sclerenchyma layers, and a deep-rooting system that enables survival in xeric psammophytic environments. Compared to related species within the genus, *A. rubtzovii* exhibits a much narrower ecological niche. For example, *Astragalus cicer* thrives in moist meadow and steppe environments and demonstrates high phenotypic plasticity [40,41], while *A. mongholicus* is distributed across semiarid grasslands and mountainous areas, with broader ecological amplitude and different drought resistance strategies [42]. The highly specialized traits of *A. rubtzovii* suggest a long evolutionary history of adaptation to its specific habitat, likely resulting in reduced ecological plasticity but enhanced performance under local conditions. Despite its distinct morphological and ecological characteristics, *A. rubtzovii* remains insufficiently studied, particularly regarding its genetic diversity, physiological responses to stress, and potential utility in conservation and restoration efforts. Owing to its unique adaptation as a xeromesophytic psammophyte inhabiting relict dune sands with intermittent moisture availability, *A. rubtzovii* holds considerable promise in terms of advancing our understanding of drought resilience and moisture-use efficiency in legumes. Moreover, its narrow geographic range and ecological specialization make it a valuable model for investigating the evolutionary processes underlying endemism, habitat specificity, and adaptation to marginal environments.

The successful assembly and characterization of the *A. rubtzovii* chloroplast genome provides valuable insights into plastid genome evolution within the Inverted Repeat-Lacking Clade (IRLC) of the Fabaceae family. The IRLC represents a large monophyletic group of papilionoid legumes characterized by the loss of one copy of the inverted repeat region [43], as evident in our assembled genome. The absence of a typical quadripartite structure in the *A. rubtzovii* chloroplast genome aligns with evolutionary patterns across the IRLC, where the loss of the IRa region is considered a strong phylogenetic signal defining this clade [44]. Our findings confirm that this structural feature is conserved in Astragalus, which is phylogenetically related to Onobrychis and Hedysarum. The size (122,984 bp) and GC content (34.6%) of the *A. rubtzovii* chloroplast genome fall within ranges observed for other IRLC species (121,020–130,561 bp and 33.6–35.1%), suggesting constraints on genome characteristics despite structural rearrangements [44,45]. We identified a relatively short 867 bp repeat region instead of the typical ~25 kb inverted repeats found in most angiosperms, reflecting significant structural evolution in the IRLC. Similar patterns have been documented in Onobrychis and other Astragalus species [44,46]. The loss of IR regions has important implications for genome stability and evolution. IRs are thought to stabilize the chloroplast genome through copy correction mechanisms, and their absence may contribute to increased genomic rearrangements and mutations in the IRLC compared to other angiosperms. Our results highlight the importance of considering phylogenetic context when assembling and annotating chloroplast genomes. Traditional expectations based on the widely conserved quadripartite arrangement may lead to misconceptions about assembly quality when working with taxa from lineages with known structural deviations. The initial uncertainty regarding our assembly’s completeness was resolved through comparison with related taxa, demonstrating the value of phylogenetically informed approaches.

### 3.3. S. nidulans

Populations of *S. nidulans* are characterized by a predominance of generative and virginile individuals, indicating a demographically stable structure in high-elevation alpine and subalpine zones. However, the low presence or absence of early developmental stages in certain sites may point to episodic recruitment driven by climatic variability or local disturbances. Morphological traits such as tomentose surfaces and dense foliage serve as effective adaptations to intense solar radiation and moisture limitation at high altitudes. Anatomically, the species displays thick-walled epidermal cells and a well-developed vascular system, reflecting structural reinforcement essential for survival in harsh mountain environments. These xeromorphic features are consistent with adaptations observed in other endemic species of Central Asia [47,48,49]. Despite this apparent reproductive success, *S. nidulans* remains insufficiently studied, with limited knowledge of its reproductive biology, genetic structure, and ecological interactions. Further research is needed to inform conservation efforts, especially given the species’ restricted range and vulnerability to anthropogenic impacts such as grazing and trampling.

To address these gaps, our study provides the first complete chloroplast genome of *S. nidulans*, a rare species endemic to Tian Shan. The genome, 152,659 bp in length, displays the typical quadripartite structure observed in angiosperms and serves as a foundational resource for evolutionary and conservation studies [50]. Given the ecological specificity and high-altitude distribution of *S. nidulans*, this endemic species is particularly vulnerable to habitat degradation and climate-driven range shifts [51,52,53]. Our analysis of its complete chloroplast genome provides a valuable basis for future studies on genetic diversity, population differentiation, and phylogenetic placement. Moreover, this plastome resource enables the development of molecular markers that can support population-level genetic assessments, helping to identify distinct or threatened populations for conservation prioritization [54]. Notably, the distinct phylogenetic placement of *S. nidulans* outside of the core *Saussurea* cluster may reflect an early divergence within the Cardueae tribe, a pattern that aligns with its unique anatomical adaptations (e.g., trichome structure and vascular patterns) and ecological specialization to montane environments. Such tools are critical for the design of effective in situ and ex situ conservation strategies tailored to the species’ limited distribution and ecological sensitivity.

### 3.4. R. wittrockii

The coenopopulations of *R. wittrockii* demonstrate demographic stability, as evidenced by the consistent recruitment of virginile and generative individuals across surveyed sites. This structure suggests the presence of functional regeneration mechanisms in natural conditions. The species’ adaptation to subalpine and forested mountain slopes is reflected in its tall growth form, extensive leaf surface area, and well-developed rhizome systems, which enable efficient resource storage and survival in heterogeneous mountain environments. Anatomical characteristics such as thick-walled epidermal cells, a well-differentiated mesophyll with prominent intercellular spaces, and structurally complex vascular bundles support mesophytic adaptation and underline the species’ ecological specialization. Beyond its ecological relevance, *R. wittrockii* also holds pharmacological potential, as a source of bioactive compounds with antioxidant, antiseptic, and anticancer properties [55,56]. These qualities make it a promising candidate for phytopharmaceutical development. However, the increasing anthropogenic pressure on its habitats and the species’ limited distribution range necessitate urgent conservation measures. Integrated approaches that combine ecological monitoring with sustainable ex situ cultivation could ensure both the preservation of natural populations and the future utilization of its medicinal potential.

The complete chloroplast genome of *R. wittrockii* yields a 141,379 bp circular genome with a typical quadripartite structure consisting of large and small single-copy regions separated by inverted repeats. Our assembly differs in size from the previously published *R. wittrockii* chloroplast genome (159,051 bp) [57], corresponding to a ~17.6 kb difference. Such difference could be attributed to variations in IR region lengths and intergenic spacers; however, the present assembly has not provided reliable evidence of their presence. This level of intraspecific variation, while substantial, has been documented in other plant species, including *Olea europaea*, with up to 13 kb differences between varieties [58], and *Populus tremula*, with variations of up to 20 kb [59]. Our annotation revealed potential gene fragments (psaA-fragment, psaB-fragment, and psbA-fragment) not reported previously, which could represent population-specific structural rearrangements or assembly artifacts. Despite size differences, our assembly contains all core angiosperm chloroplast genes with proper intron positioning, including the correctly trans-spliced *rps12* gene, confirming the assembly’s completeness and quality. The annotation of *rpl23* as a functional gene in our assembly versus its classification as a pseudogene in the previous study [57] further highlights potentially significant variation between populations. The observed genomic differences likely reflect natural intraspecific diversity within this threatened endemic species. Such variation can result from geographical isolation, local adaptation, and founder effects—which are particularly relevant for species with fragmented distributions across mountain ranges like the Ile Alatau and Uzynkara ranges [60]. From a conservation perspective, documenting genetic variation within threatened species is crucial, as recent research emphasizes the preservation of intraspecific diversity to maintain evolutionary potential and environmental resilience [61].

Collectively, these findings underscore the necessity of species-specific conservation measures. For *T. kok-saghyz* and *S. nidulans*, regulating overgrazing and trampling is critical to maintaining population viability. Conservation of *A. rubtzovii* should focus on protecting its narrow psammophytic habitat from degradation, while *R. wittrockii* would benefit from continued monitoring to ensure the persistence of stable recruitment. Additionally, all four species exhibit some capacity for vegetative reproduction, including root sprouting, rhizome or root fragmentation, clump division, and stem rooting. This capacity may enhance persistence and facilitate local recovery following disturbance, partly mitigating the limitations associated with low seed recruitment. Integrating morphological, ecological, and genomic data offers a robust framework for identifying vulnerable populations and developing effective long-term conservation strategies for Kazakhstan’s endemic flora.

## 4. Materials and Methods

### 4.1. Plant Material

Sample collection, field surveys, and stationary observations were carried out on the Ile Alatau and Ketmen (Uzynkara) ridges of the Northern Tian Shan (Figure 5). On the Uzynkara Ridge, *T. kok-saghyz* and *A. rubtzovii* were identified at elevations of approximately 1920 masl and 1910 masl, respectively. *R. wittrockii* and *S. nidulans* were documented on both ridges, with populations observed at 2263 masl and 3260 masl on the Ile Alatau Ridge and at 2105 masl and 3003 masl on the Uzynkara Ridge. These data provide important insights into the altitudinal distribution and habitat preferences of rare and endemic plant species within distinct mountain systems of southeastern Kazakhstan. The species composition of collected plants was identified using regional floristic keys and illustrated guides [62,63,64], with genus classification following S.A. Abdullina [65] and species nomenclature verified according to S.K. Cherepanov [66].

### 4.2. Geobotanical Survey and Population Structure

Ecological and population structures were assessed using route-reconnaissance surveys and standard geobotanical methods [67,68]. Species abundance in plant communities was estimated using the Drude scale. Age classes were defined following Rabotnov [69] and Gatsuk et al. [70] as follows: juvenile (young plants with small root systems and no reproductive structures), immature (larger, not yet reproductive vegetative plants), virginile or vegetative adult (mature plants capable of flowering but not flowering during the observation period), and generative (flowering or fruiting individuals). For larger populations, three 1 × 1 m plots were randomly established per site, whereas all individuals were recorded in smaller populations. Vegetation cover, life-form classification [71,72], topography, and GPS coordinates were recorded for each plot. Statistical analyses were performed using SPSS (v20.0) with an independent t-test at a 5% significance level.

### 4.3. Morphological and Anatomical Studies

Morpho-anatomical analysis was conducted to characterize root, stem, and leaf structures across species. Plant material was fixed in a 1:1:1 solution of ethanol:glycerin:water and sectioned using an OL-ZSO freezing microtome (Inmedprom, Yaroslavl, Russia). Tissues were examined under an MC 300 microscope (Micros, Austria) with a CAM V400/1.3M camera (jProbe, Japan). Measurements were performed using an MOV-1-15 eyepiece micrometer at ×10 and ×40 magnification. Stem samples were treated with 5% NaOH to remove epidermis and assess underlying tissue structures. Anatomical traits were described following standard protocols in botanical microtechnology [73,74,75].

### 4.4. DNA Extraction and Sequencing

DNA was extracted from 100 mg of leaf material using a modified CTAB protocol. The quality of the DNA was assessed using Nanodrop and Qubit. Whole-genome sequencing was performed using the SurfSeq 5000 (Genemind, China) platform. Raw reads were quality-filtered using fastp v0.23.2 [76]. Chloroplast genomes were assembled de novo using GetOrganelle v1.7.5 with the seed_plant parameter [77]. GetOrganelle employs an iterative approach that first identifies reads of organellar origin through mapping to reference sequences, followed by de novo assembly using SPAdes. For all four species, we obtained complete and circular chloroplast genome assemblies in a single scaffold without further scaffolding steps required. The initial assemblies were polished using Pilon v1.24 to correct potential sequencing errors [78]. Cleaned reads were mapped to the draft assemblies using BWA-MEM v0.7.17 [79], and Pilon was run with default parameters to improve base accuracy and resolve small indels. The polished chloroplast genome assemblies were annotated using GeSeq [80] with a multi-evidence approach: BLAT searches against reference proteins (blatX, 60% identity threshold) and nucleotides (blatN, 75% identity threshold), complemented by tRNAscan-SE v2.0.7 for tRNA prediction [81].

Circular gene maps of the chloroplast genomes were generated using OGDRAW v1.3.1, which provides a visual representation of gene organization, direction, and functional categorization across the genome. The chloroplast genomes were depicted as circular molecules with genes color-coded according to their functional categories. For the phylogenetic analysis, complete reference sequences of plastid genomes were retrieved from the NCBI Nucleotide database: 55 species of the Cichoreae tribe of the Asteraceae family for *T. kok-saghyz*; 67 species of the Galegeae tribe of the Fabaceae family for *A. rubtzovii*; 62 species of Cardueae tribe of the Asteraceae family for *S. nidulans*; and 59 species of the Polygonoideae subfamily of the Polygonaceae family for *R. wittrockii.* Each dataset included all available species of the corresponding genus and 1–3 species representing all other genera. The assembled sequences of the plastid genomes of *T. kok-saghyz*, *A. rubtzovii*, *S. nidulans*, and *R. wittrockii* were aligned against the corresponding set of sequences using the NCBI nBLAST tool [82] with the MegaBLAST settings (optimized for similar sequences). After that, the ORPA tool was used to prepare aligned sequences for the phylogenetic inference [83]. The neighbor-joining tree was calculated using MEGA 11 software (version 11.0.11) with the following settings: Tamura–Nei substitution model, uniform rates among sites, pairwise deletion of gaps/missing sites, and 1000 bootstrap replicates [84]. Finally, the phylogenetic trees were visualized using FigTree v.1.4.4 [85].

## 5. Conclusions

This integrative study of *T. kok-saghyz*, *A. rubtzovii*, *S. nidulans*, and *R. wittrockii* provides the first comprehensive assessment of their ecological, anatomical, and genomic characteristics within the Northern Tian Shan. By combining chloroplast genome sequencing with population structure and morphological analyses, we uncovered distinct habitat adaptations and genetic profiles that reflect the evolutionary trajectories and conservation needs of these rare and endemic taxa. Despite their capacity for local adaptation, all four species face increasing threats from anthropogenic pressures such as overgrazing, habitat fragmentation, and land-use change. The genomic data, particularly the identification of IR-lacking plastomes in *A. rubtzovii* and population-specific genome variation in *R. wittrockii*, underscore the need for phylogenetically informed conservation approaches. Our findings highlight the urgency of implementing targeted, species-specific conservation strategies, including habitat protection, long-term population monitoring, and the development of molecular markers for future genetic studies. Together, these species represent a gradient of ecological strategies and chloroplast genomic features, from mesophytic taprooted herbs to xeromesophytic legumes, each requiring conservation approaches tailored to their unique biology and environmental context. This research contributes a critical foundation for preserving the genetic diversity and ecological integrity of Kazakhstan’s endemic flora amid accelerating environmental change.

## Figures and Tables

**Figure 1 plants-14-02305-f001:**
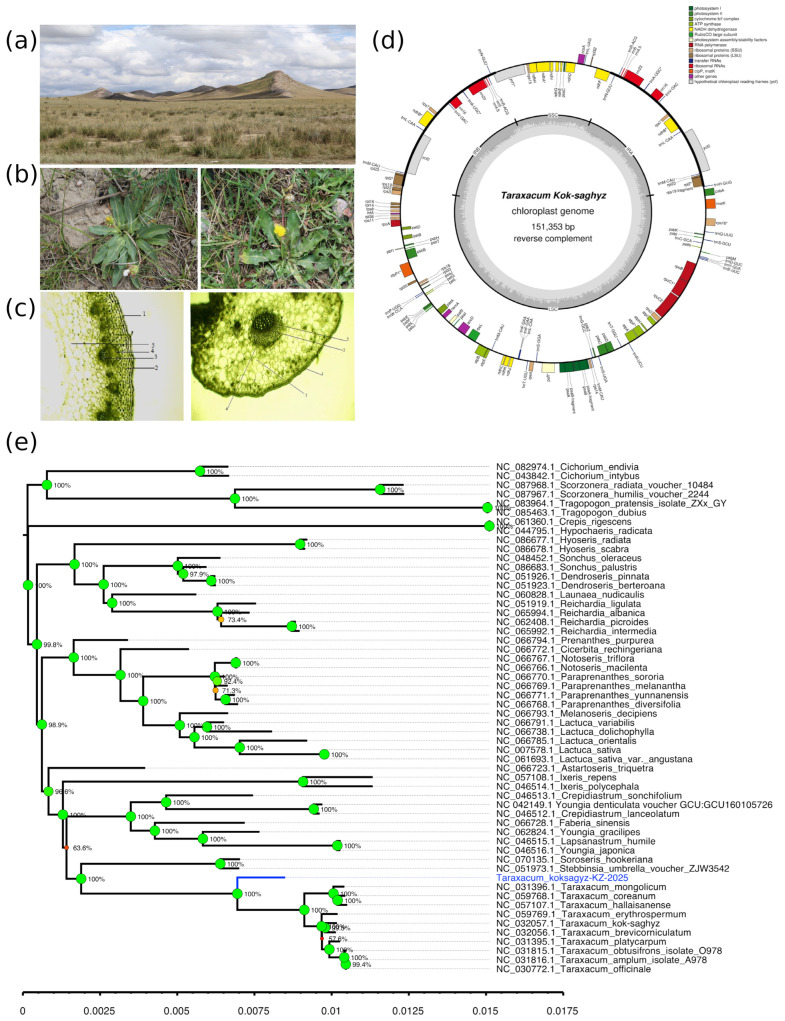
Morphological, anatomical, and chloroplast genomic features of *Taraxacum kok-saghyz*. (**a**) Habitat view in the intermountain valleys of the Northern Tian Shan (Kegen region). (**b**) Rosette morphology and flowering individuals in natural conditions. (**c**) Cross-section of stem and root tissues under light microscopy (×10): 1—epidermis; 2—primary cell wall; 3—vascular bundle; 4—sclerenchyma; 5—core parenchyma. (**d**) Annotated chloroplast genome map (151,353 bp; reverse complement) with protein-coding genes, tRNAs, rRNAs, and hypothetical reading frames. (**e**) Neighbor-joining tree of chloroplast genome of *T. kok-saghyz* with the species of the Cichoreae tribe.

**Figure 2 plants-14-02305-f002:**
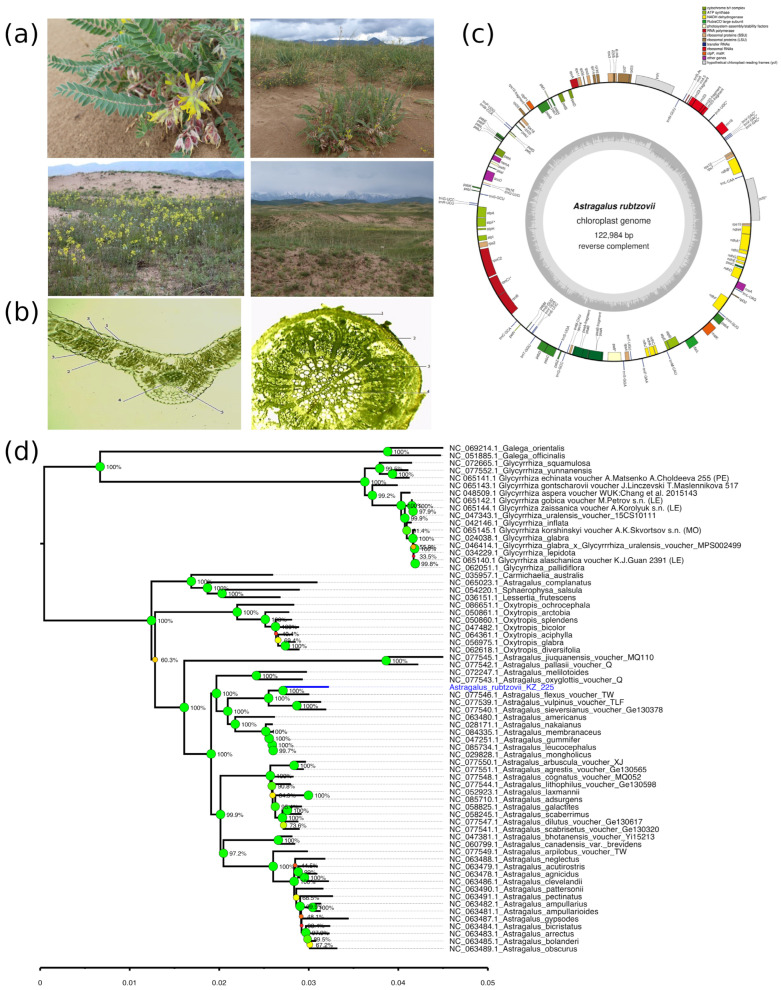
Morphological, anatomical, and genomic features of *Astragalus rubtzovii*. (**a**) Habitat and vegetative morphology on the relict dune sands of Kum-Tekey, Uzynkara Ridge, including flowering individuals and plant community structure. (**b**) Anatomical structure under light microscopy (×10): (**left**)—transverse section of the leaf showing (1) the upper epidermis, (2) lower epidermis, (3) columnar mesophyll, (4) vascular bundle, and (5) sclerenchyma; (**right**)—transverse section of the root showing (1) the exoderm, (2) primary sheath, (3) vascular bundle, and (4) xylem. (**c**) Annotated circular map of the chloroplast genome (122,984 bp; reverse complement), showing gene functional groups and GC content variation. (**d**) Neighbor-joining tree of chloroplast genome of *A. rubtzovii* with the species of the Galegeae tribe.

**Figure 3 plants-14-02305-f003:**
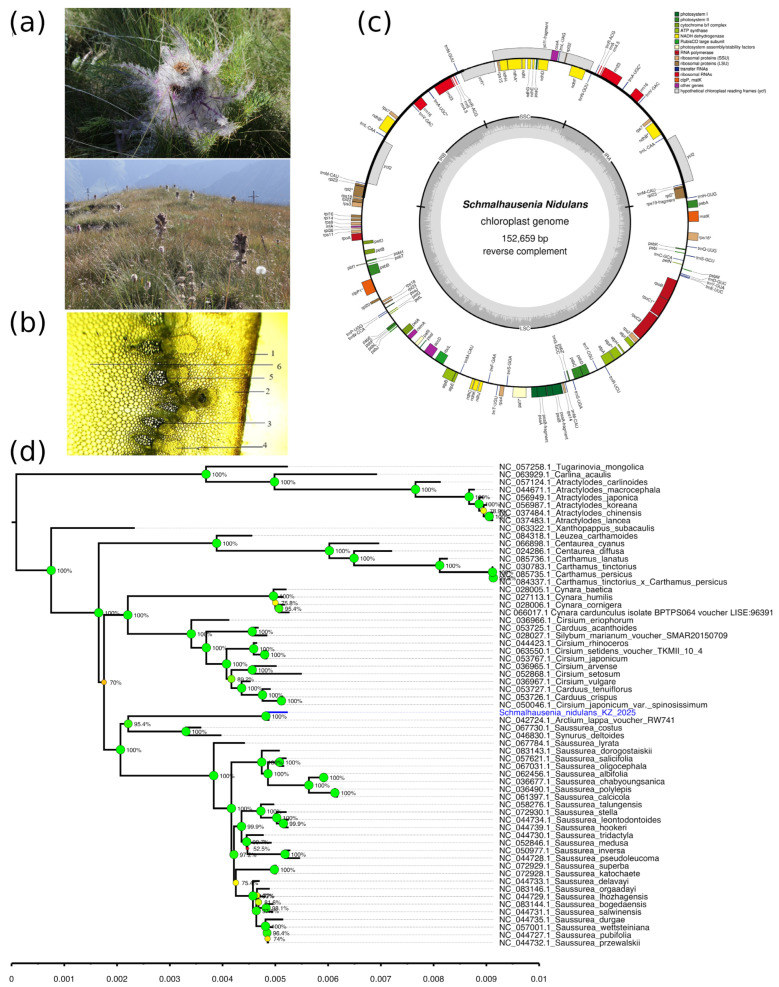
Morphological, anatomical, and chloroplast genomic features of *Schmalhausenia nidulans*. (**a**) Close-up view of a flowering individual in its natural habitat. (**b**) Transverse section of stem tissue under light microscopy (×10): 1—epidermis; 2—loose parenchyma; 3—vascular bundle; 4—sclerenchyma; 5—xylem; 6—vascular parenchyma. (**c**) Annotated circular map of the chloroplast genome (152,659 bp; reverse complement) with protein-coding genes, tRNAs, rRNAs, and hypothetical open reading frames. (**d**) Neighbor-joining tree of the chloroplast genome of *S. nidulans* with the species of the Cardueae tribe.

**Figure 4 plants-14-02305-f004:**
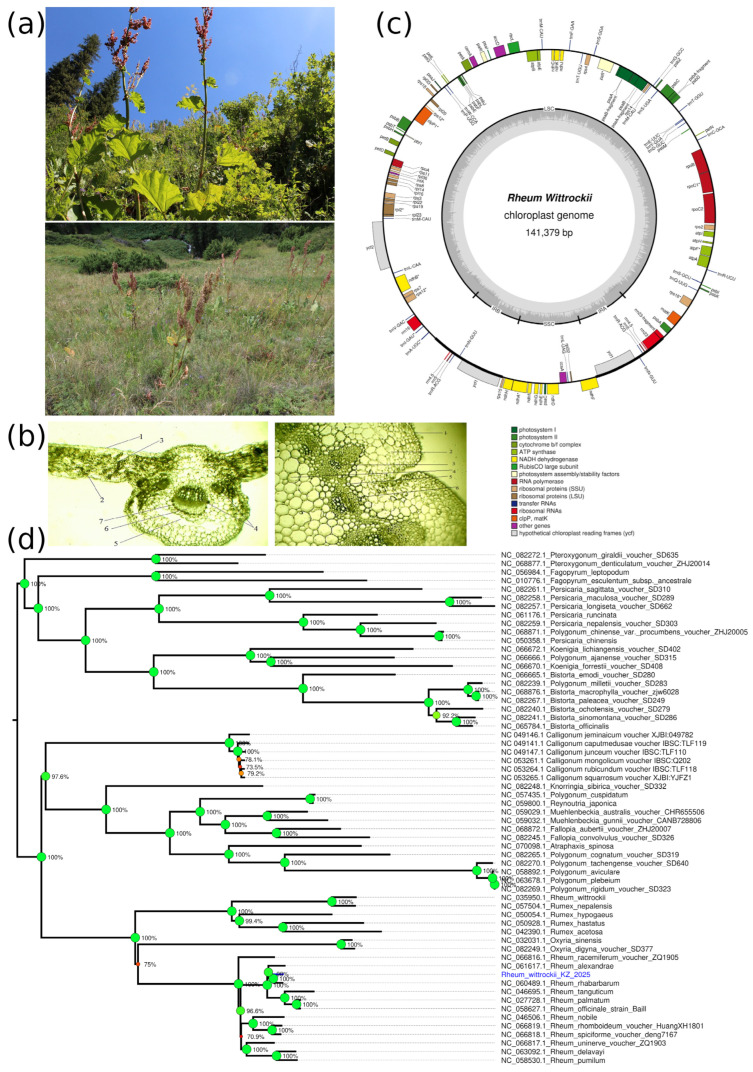
Morphological, anatomical, and chloroplast genomic features of *Rheum wittrockii*. (**a**) Habitat views in the Northern Tian Shan: (**top**) flowering individuals under open canopy; (**bottom**) population-level view in montane grassland. (**b**) Anatomical structure of leaf and stem under light microscopy (×10): (**left**) transverse section of the leaf (1—upper epidermis; 2—lower epidermis; 3—columnar mesophyll; 4—vascular bundle; 5—aerial streak; 6—vascular bundle of sclerenchyma; 7—idioblast) (**right**) transverse section of the stem (1—epidermis; 2—primary sheath; 3—vascular bundle; 4—core parenchyma; 5—endoderm; 6—sclerenchyma). (**c**) Annotated circular map of the chloroplast genome (141,379 bp), showing protein-coding genes, tRNAs, rRNAs, and hypothetical open reading frames. (**d**) Neighbor-joining tree of chloroplast genome of *R. wittrockii* with the species of the Polygonoideae subfamily.

**Figure 5 plants-14-02305-f005:**
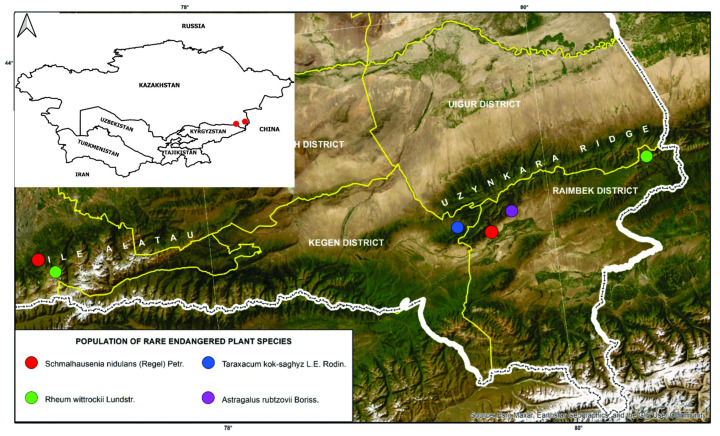
Geographic distribution of rare and threatened plant species in the Northern Tian Shan. The map shows the locations of documented populations of *S. nidulans* (red), *T. kok-saghyz* (blue), *R. wittrockii* (green), and *A. rubtzovii* (purple) across the Ile Alatau and Ketmen (Uzynkara) ridges in southeastern Kazakhstan. District boundaries (yellow) and international borders (white) are indicated. The inset highlights the study region within Central Asia.

**Table 1 plants-14-02305-t001:** Comparative morphometric traits of *Taraxacum kok-saghyz* across developmental stages in two natural populations.

Population	Age Group	Number of Individuals	Leaf Length (cm) ± SD	Plant Height (cm) ± SD
Pop. 1	Juvenile (j)	45	1.13 ± 0.32	1.37 ± 0.42
	Immature (i)	41	1.80 ± 0.26	1.80 ± 0.14
	Virginile (v)	64	3.47 ± 0.45	2.67 ± 0.25
	Generative (g)	30	8.33 ± 0.58	11.00 ± 1.00
Pop. 2	Juvenile (j)	30	1.93 ± 0.21	0.67 ± 0.15
	Immature (i)	39	2.60 ± 0.17	1.85 ± 0.07
	Virginile (v)	52	4.30 ± 0.53	3.20 ± 0.36
	Generative (g)	61	7.00 ± 3.61	12.00 ± 2.00

Statistically significant differences were observed at specific developmental stages (*p* < 0.05).

**Table 2 plants-14-02305-t002:** Morphometric characteristics of *Astragalus rubtzovii* at different developmental stages.

Population	Age Group	Number of Individuals	Leaf Length (cm) ± SD	Plant Height (cm) ± SD
Population 1	Juvenile (j)	30	1.60 ± 0.66	9.00 ± 10.00
	Immature (i)	610	1.83 ± 0.15	13.33 ± 15.28
	Virginile (v)	949	1.97 ± 0.06	25.00 ± 30.00
	Generative (g)	730	2.10 ± 0.36	42.33 ± 68.07

Statistically significant differences were observed at specific developmental stages (*p* < 0.05).

**Table 3 plants-14-02305-t003:** Morphometric characteristics of *Schmalhausenia nidulans* at different developmental stages across two populations.

Population	Age Group	Number of Individuals	Leaf Length (cm) ± SD	Plant Height (cm) ± SD
Pop. 1	Juvenile (j)	65	6.67 ± 2.08	8.00 ± 1.00
	Immature (i)	31	10.33 ± 1.53	20.00 ± 5.00
	Virginile (v)	154	18.33 ± 1.53	23.67 ± 6.03
	Generative (g)	189	31.33 ± 3.21	29.67 ± 2.52
Pop. 2	Juvenile (j)	20	5.67 ± 1.53	11.33 ± 1.53
	Immature (i)	29	10.67 ± 2.08	12.33 ± 1.53
	Virginile (v)	33	16.17 ± 1.04	22.33 ± 2.52
	Generative (g)	41	26.00 ± 3.61	30.67 ± 4.04

Statistically significant differences were observed at specific developmental stages (*p* < 0.05).

**Table 4 plants-14-02305-t004:** Morphometric characteristics of *Rheum wittrockii* at different developmental stages across two populations.

Population	Age Group	Number of Individuals	Leaf Length (cm) ± SD	Plant Height (cm) ± SD
Pop. 1	Juvenile (j)	35	8.83 ± 0.76	15.67 ± 1.15
	Immature (i)	47	14.17 ± 1.20	30.00 ± 5.00
	Virginile (v)	74	17.83 ± 1.26	47.33 ± 7.51
	Generative (g)	29	31.17 ± 3.40	83.33 ± 7.64
Pop. 2	Juvenile (j)	29	6.67 ± 1.53	17.33 ± 2.52
	Immature (i)	33	11.33 ± 1.53	23.33 ± 1.53
	Virginile (v)	63	19.33 ± 2.08	40.67 ± 9.29
	Generative (g)	38	30.67 ± 4.04	78.33 ± 7.64

Statistically significant differences were observed at specific developmental stages (*p* < 0.05).

## Data Availability

The datasets generated and analyzed during the current study are available in the Open Science Framework (OSF) repository: DOI 10.17605/OSF.IO/6AU8N.

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
