# Peer review of "Insights into Biological and Ecological Features of Four Rare and Endemic Plants from the Northern Tian Shan (Kazakhstan)"

_plants, 2025, doi:10.3390/plants14152305_

Round 1
Reviewer 1 Report
Comments and Suggestions for Authors
The manuscript brings many new and interesting reports species of conservation interest both in Kazakhstan and internationally.
However, it should be improved.
The term virginal is not common. You do not mention according to which reference You studied age structure. Is virginal equal to sterile plant, non-generative mature, pre-mature. Please elaborate. Please, use the appropriate term in accordance with recent literature instead of colloquial VIRGIN. Age structure needs reference support.
Please, elaborate distinction between virginal-juvenile, juvenile - immature. It has to be clear to all readership.
In some places, I have impression You use juvenile for pre-mature, immature and in other for seedling. Please, make it consistent.
Also elaborate and make these issues clear to readers.
Why did You calculated number of leaves per m2, and not number of leaves per plant? Why not number of plants per m2?
In Figures 1-4, the C is rather invisible and should be bigger.
Latin names are not properly written throughout the text. Please, use the authorities where the species firstly is mentioned in the text for whatever mentioned species.
Italic letter must be applied for genus and tribe as well, and the kok-saghyz must be written consistently (not koksaghyz). Similarly, Wittrockii should be wittrockii.
The full species names, not abbreviations must be used in figure labels, at the beginning of the sentences, and in subtitle. Subtitle should be more informative than only species names.
What is community height?
line 372-374 is not true. Chloroplast DNA is rather conserved. Can You support this statements with the reference?
line 383-388 is superfluous. You can delete it or elaborate in details.
You often use endemic, but You have not elaborated endemic to what region, areas or floras.
You have not reported or mentioned any reproductive potential of theses species vegetativelly. Any possibility of vegetative spread or survival?
Endangered is IUCN threat category, so if this is not the case, please use threaten and derivatives (e.g. key words).
You are mentioning soil type and phytocoenology, but these are not included into this manuscript / neither presented nor discussed.
Please, try to improve the manuscript according to comments.
I think the title should be change to accommodate better Your results to:
Insights into biological and ecological features of four rare and endemic plants from the Northern Tien Shan (Kazakhstan).
Author Response
Dear Reviewer,
We sincerely thank you for your detailed and thoughtful comments on our manuscript “Insights into biological and ecological features of four rare and endemic plants from the Northern Tien Shan (Kazakhstan).” We greatly appreciate the time and care you devoted to helping improve our work.
We have carefully revised the manuscript to address all of your points. Below we provide a point-by-point response, indicating the specific changes made.
Reviewer Comment: The term virginal is not common. You do not mention according to which reference You studied age structure. Is virginal equal to sterile plant, non-generative mature, pre-mature. Please elaborate. Please, use the appropriate term in accordance with recent literature instead of colloquial VIRGIN. Age structure needs reference support. Please, elaborate distinction between virginal-juvenile, juvenile - immature. It has to be clear to all readership. In some places, I have impression You use juvenile for pre-mature, immature and in other for seedling. Please, make it consistent. Also elaborate and make these issues clear to readers.
Response: We agree that the term “virginal” can be unclear or ambiguous to readers unfamiliar with its use in Russian and Central Asian geobotanical literature, where it traditionally designates a mature vegetative, pre-reproductive phase. To improve clarity, we have carefully revised the manuscript as follows:
- We replaced “virginal” with “virginile (mature vegetative)” throughout the text to align with terminology used in ecological demography studies of perennial herbs (e.g., Rabotnov 1969; Gatsuk et al. 1980).
- We added an explicit definition of age classes in the Materials and Methods section:
“Age classes were defined following Rabotnov [69] and Gatsuk et al. [70] as: juvenile (young plants with small root systems and no reproductive structures), immature (larger vegetative plants not yet reproductive), virginile or vegetative adult (mature plants capable of flowering but not flowering during the observation period), and generative (flowering or fruiting individuals).”
- We ensured consistent use of these terms throughout the manuscript, avoiding overlap or confusion between juvenile, immature, and seedling stages.
- We included supporting references for the age classification system (Rabotnov, 1969; Gatsuk et al., 1980), which is widely used in Central Asian and Russian plant population ecology.
These revisions aim to make our age-structure terminology fully transparent and accessible to an international readership.
Reviewer Comment: Why did you calculate number of leaves per m², and not number of leaves per plant? Why not number of plants per m²?
Response: Thank you for this helpful comment. We agree that the description was unclear and potentially misleading. To improve clarity and consistency with our actual measurements, we have revised the text to remove that part and ensure that only directly relevant morphometric traits per individual plant are reported in the manuscript.
Reviewer Comment: In Figures 1–4, the C is rather invisible and should be bigger.
Response: We have carefully revised the figure panels to improve clarity and readability. Specifically:
- We increased the size of the panel labels, with particular attention to panel C, which was previously less visible. The larger, bolder labels now ensure clear identification of all subfigures.
- We added the phylogenetic trees as new panel D in each figure to better illustrate the phylogenetic placement of the studied species based on their complete chloroplast genome sequences.
- We updated all figure captions to clearly describe the newly included phylogenetic tree panels.
Reviewer Comment:Latin names are not properly written throughout the text. Please, use the authorities where the species firstly is mentioned in the text for whatever mentioned species. Italic letter must be applied for genus and tribe as well, and the kok-saghyz must be written consistently (not koksaghyz). Similarly, Wittrockii should be wittrockii. The full species names, not abbreviations must be used in figure labels, at the beginning of the sentences, and in subtitle. Subtitle should be more informative than only species names.
Response: We carefully revised the manuscript to address all of these points:
- We ensured that full Latin names with authorities are provided at their first mention in the Abstract and Introduction.
- We standardized spelling throughout the text, consistently using kok-saghyz and wittrockii in lowercase.
- We applied italics consistently to all genus and species names, as well as tribe names where appropriate.
- For all figure captions and table titles, we replaced abbreviations with full species names in italics and improved the subtitles to be more informative and descriptive.
Reviewer Comment: What is community height?
Response: We revised the Results section to clarify this phrase. It now describes the stands as having 45–70% total vegetation cover with dominant species reaching heights of up to 80–90 cm.
Reviewer Comment: Line 372–374 is not true. Chloroplast DNA is rather conserved. Can you support this statement with the reference?
Response: We agree that chloroplast DNA is generally conserved in structure and gene content across angiosperms. We have revised this sentence to acknowledge its conserved nature while noting that variable regions can still provide useful markers for conservation genetics. We also added appropriate references to support this statement.
Reviewer Comment: Line 383–388 is superfluous. You can delete it or elaborate in detail.
Response: We agreed with the comment and have deleted lines 383–388 from the manuscript.
Reviewer Comment: You often use endemic, but you have not elaborated endemic to what region, areas or floras.
Response: We clarified the regional context on lines 65–67 of the Introduction, specifying that these species are endemic to the Northern Tian Shan of Kazakhstan.
Reviewer Comment: You have not reported or mentioned any reproductive potential of these species vegetatively. Any possibility of vegetative spread or survival?
Response: We have now addressed this point in the Discussion section, adding a new sentence that describes the known capacity of all four species for vegetative reproduction, including root sprouting, rhizome or root fragmentation, clump division, and stem rooting.
Reviewer Comment: Endangered is IUCN threat category, so if this is not the case, please use threatened and derivatives (e.g. key words).
Response: We have replaced all instances of “endangered” with “threatened” throughout the manuscript to ensure accurate and appropriate terminology.
Reviewer Comment: You are mentioning soil type and phytocoenology, but these are not included into this manuscript / neither presented nor discussed.
Response: We agree with this point and have removed references to soil type and phytocoenology from the Methods section to ensure consistency with the presented data.
Reviewer Comment: I think the title should be changed to accommodate better your results to: Insights into biological and ecological features of four rare and endemic plants from the Northern Tien Shan (Kazakhstan).
Response: Thank you for this helpful suggestion. We agree and have updated the title to: Insights into biological and ecological features of four rare and endemic plants from the Northern Tien Shan (Kazakhstan).
We thank you again for your careful and constructive review, which has greatly improved the clarity and quality of our manuscript.

Reviewer 2 Report
Comments and Suggestions for Authors
Authors characterized the populations of rare species in Northern Tien Shan, Kazakhstan. Authors conducted geobotanical surveys, population and structural analyses and sequencing of chloroplast genomes, and described the basic information of its rarity and adaptation to the alpine grassland environments. Although the data were limited and the study was preliminary, habitat surveys, descriptions of population structures, and inner structural analyses were relatively well documented (Ideally, authors should conduct using more individuals to confirm the generality of the described structural features). On the other hand, analyses using chloroplast genome data had severe problems. Descriptions of genome structures and gene contents might be no large problems, but results of phylogenetic analyses are quite strange. All 4 trees have the same strange points: branches of target species (and some accessions accompanied with the target species) were too long and the values of distances were too large (except for the Rheum wittrockii but see below). Descriptions of phylogenetic analyses were also insufficient (e.g., which evolutionary models were selected for each analysis, log-likelihood of the selected tree). Phylogenetic relationships of the target taxa should be redescribed after solving the following problems.
(1) In the case of Taraxacum, I roughly compared with the accessions closely related to Taraxanum_kok-saghyz and those distantly related to the species. I then found that two data closely related to Taraxanum_kok-saghyz (ON641329 and OL875302) were registered as the complementary sequences to the other chloroplast genomes of Traxacum (and probably other asteracean cp gemone data). That might be the cause of the extreme long blanche of clade with Taraxanum_kok-saghyz if the authors use those data as it is directly from the database. For the other two genera with the similar trends (Astragalus rubtzovii and Schmalhausenia nidulans), the same problems might be present in the analyses.
(2) For the case of Rheum wittrockii, the branch to the species was still long but the distance was not so large comparing with those of other 3 genera. Authors should check the accuracy of sequencing. There are no such data (e.g. sequencing depth) to judge the accuracy of sequencing analyses, so the authors also should add the descriptions of primary sequencing results.
(3) We already can use the registered sequences of cp genome for Taraxanum_kok-saghyz (NC_032057=KX198560), so the authors should include the data for the analysis to compare the genome sequences of the same species with the different origin. Chloroplast genome data of Taraxacum have been still limited despite the plenty of species in the world, and some data did not appear in the analysis (e.g., Taraxacum amplum (NC_031816), T. obtusifrons (NC_031815), T. albidum (LC790150), T. brevicorniculatum (NC_032056)). Authors should put all chloroplast genome data for the phylogenatic analysis.
(4) There were no obvious outgroup(s) in the phylogenetic analysis of Astragalus. Authors should put data of some other genera (e.g., Oxytropis) for outgroup(s). As the case of Taraxacum, some chloroplast genome data of Astragalus (NC_085710, OR491697, OR491699, and so on) did not appear in the analysis.
(5) Schmalhausenia nidulans was currently treated as Olgaea nidulans (see POWO). But phylogenetic study of Chinese asteracean plants rejected the congeneric habit of Schmalhausenia nidulans and Olgaea spp. (Fu et al. 2016, Journal of Systematics and Evolution 54: 273-467). Although I did not check chloroplast genome data throughly, authors should check the presence of other data of chloroplast genome for Cardueae. At least chloroplast genome of Xanthopappus subacaulis (MT643189) was already published and authors should include. As the case of Astragalus, outgroup(s) of the analysis was not obviously indicated. From the results of general classification of Asteraceae, Inura belonging to different tribe was the appropriate aoutgroup, but the position seems to be completely inside of the tribe Cardueae. Authors should check the data and select appropriate outgroup(s) for the analysis.
Author Response
Dear Reviewer,
Thank you very much for your detailed and constructive comments. We appreciate the time and care you took to review our phylogenetic analyses. Below, we address each of your concerns.
Response (1-5): We have completely re-evaluated the phylogenetic analysis in the article. Whereas the previous results have been based on the direct alignment of complete plastid genome sequences which leads to mistakes due to the structural variation of the sequences, the present results rely on the comparison of the homologous sites using local alignment (BLAST) and recently developed and published ORPA tool. The newly obtained result complies with the known taxonomy, have high bootstrap support and thus can be considered reliable. All trees included the representative species of the related genera as the outgrups. All the details of used phylogenetic models have been added to the description of the methods.
Response (1,3): We have added all available reference Taraxacum species including T. kok-saghyz. The discrepancy between the reference sequence of T. kok-saghyz and ours, as well as its separation of other species, have been mentioned in the discussion.
Response (4): We added related genera for R. wittrockii including all available reference of genus Rheum. Because the limitations of BLAST alignment for the input data size (10,000,000 bases total) we have included only reference sequences (prefix NC) and have been able to use the limited numbers of species outside the target genera.
Response (2): The newly applied method helped to ger rid of the branch length artifact and obtain more reliable structure.
Response (5): We have re-evaluated S. nidulans as the synonym of Arctium eriophorum, referring the NCBI Taxonomy database. Thus, the phylgenetic clustering of S. nidulans with A. lappa confirms the correctness of the newly applied method.

Reviewer 3 Report
Comments and Suggestions for Authors
Title: Integrative analysis of genetic and ecological differentiation in rare endemic plant species of the Northern Tien Shan
Well written manuscript that summarize new botanical and genetic findings of three rare endemic plants. Material and methods section is written in a fully reproducible mode.
My only minor comment is, the figure of chloroplast genomes should be larger and more visible.
Author Response
Dear Reviewer,
Thank you very much for your positive evaluation of our manuscript and for highlighting its clarity and reproducibility. We appreciate your helpful suggestion regarding the chloroplast genome figure. In response, we have increased its size and improved its resolution to ensure greater visibility and ease of interpretation for readers.
Thank you again for your valuable feedback.

Reviewer 4 Report
Comments and Suggestions for Authors
Review of “Integrative analysis of genetic and ecological differentiation in 2 rare endemic plant species of the Northern Tien Shan”
This manuscript presents detailed population structural, morphological and chloroplast genome sequencing studies of four endemic species (Taraxacum kok-saghyz, Astragalus rubtzovii, Schmalhausenia nidulans and Rheum wittrockii) in the Northern Tien Shan. The authors have done a rigorous analysis of the data and present all important details clearly. The manuscript is appropriate for publication in Plants.
Below are some detailed comments:
Ln. 3: Tien Shan or Tian Shan, please use only one form of spelling.
Ln. 103: “masl” (meter above sea level) needs to be defined when it appears for the first time.
Lns. 349-360: are rRNA genes present in duplicated copies in inverted repeats (IR) in R. wittrockii? The chloroplast genome map shown on p. 9 shows no IR, or the map is in such a low resolution that it is difficult to see all rRNA genes in two copies distributed symmetrically on both sides of the single copy regions. The entry (NC_035950) in GenBank for this species submitted by a different research group shows that the genome has an IR.
Lns.: 515-517: you say that R. wittrockii, but the map you show on p. 9 does not reflect this point. Please clarify.
Some Latin names of genera are not italicized.
Author Response
Dear Reviewer,
We sincerely thank you for your detailed and thoughtful comments on our manuscript “Insights into biological and ecological features of four rare and endemic plants from the Northern Tien Shan (Kazakhstan).” We greatly appreciate the time and care you devoted to helping improve our work.
We have carefully revised the manuscript to address all of your points. Below we provide a point-by-point response, indicating the specific changes made.
Reviewer comment: Ln. 3: Tien Shan or Tian Shan, please use only one form of spelling.
Response: We have standardized the spelling of “Tian Shan” throughout the manuscript for consistency.
Reviewer comment: Ln. 103: “masl” (meter above sea level) needs to be defined when it appears for the first time.
Response: We have defined the abbreviation “masl” at its first occurrence as “meters above sea level (masl).”
Reviewer comment: Lns. 349-360: are rRNA genes present in duplicated copies in inverted repeats (IR) in R. wittrockii? The chloroplast genome map shown on p. 9 shows no IR, or the map is in such a low resolution that it is difficult to see all rRNA genes in two copies distributed symmetrically on both sides of the single copy regions. The entry (NC_035950) in GenBank for this species submitted by a different research group shows that the genome has an IR.
Response: After re-evaluation of the obtained plastid genome assembly, we have concluded that the presence of IR has not been reliably supported. The indication of this fact has been added to the text of the article.
Reviewer comment: Lns.: 515-517: you say that R. wittrockii, but the map you show on p. 9 does not reflect this point. Please clarify.
Response: We agree that the map on p. 9 reflects our own assembly, which has the ~17.6 kb shorter size compared to the published R. wittrockii chloroplast genome. As such, this map does not show the difference directly, but instead illustrates our reconstructed genome structure with its own measured size.
Reviewer comment: Some Latin names of genera are not italicized.
Response: We have checked the entire manuscript and ensured that all Latin genus and species names are correctly italicized.

Round 2
Reviewer 1 Report
Comments and Suggestions for Authors
Thank You for Your reply to my comments.